# Research on equipment fault diagnosis model based on gan and inverse PINN: Solutions for data imbalance and rare faults

Jian Deng[1], Zheng Cheng[1]*, Aiming Gu[2], Shibohua Zhang[3]

1 School of Economics and Management, Leshan Normal University, Leshan, China, 2 Jiaxing First Hospital, Jiaxing, China, 3 School of economics and management, Xi'an University of Technology, Xi'an, China

* chengzheng@lsnu.edu.cn

## Abstract

In the field of medical imaging equipment, fault diagnosis plays a vital role in guaranteeing stable operation and prolonging service life. Traditional diagnostic approaches, though, are confronted with issues like intricate fault modes, as well as scarce and imbalanced data. This paper puts forward a fault diagnosis model integrating digital twin technology and Inverse Physics - Informed Neural Networks (Inverse PINN).The practical significance of this research lies in its potential to revolutionize the engineering aspects of medical imaging equipment management. By constructing a physical model of equipment operation and leveraging inverse PINN to deal with imbalanced datasets, the model can accurately identify and predict potential faults. This not only optimizes the full lifecycle management of the equipment but also has the potential to reduce maintenance costs, improve equipment availability, and enhance the overall efficiency of medical imaging services.Experimental results show that the proposed model outperforms in fault detection and prediction for medical imaging equipment, especially making breakthroughs in data generation and fault detection accuracy. Finally, the paper discusses the model's limitations and future development directions.

## Introduction

Medical imaging equipment forms a core part of modern healthcare systems, and its reliability and accuracy directly impact patient diagnosis and treatment outcomes. Due to the complexity of the equipment and its prolonged use under high-load operations, frequent equipment failures result in high maintenance costs, long downtimes, and interruptions to medical services. Therefore, developing an accurate and efficient fault diagnosis system is essential to improving the utilization of medical imaging equipment and ensuring the quality of healthcare services.

**Data availability statement:** All relevant data are within the paper and its Supporting Information files.

**Funding:** This work was supported in part by the Scientific Research Program at Leshan Normal University, the Natural Science Key Foundation of Sichuan under grant number 2023YFG0114, the National Natural Science Foundation of China under grant number 61902039. The funders had no role in study design, data collection and analysis, decision to publish, or preparation of the manuscript.

**Competing interests:** The authors have declared that no competing interests exist.

Traditional fault diagnosis methods rely on historical fault data or monitoring under specific conditions, leading to challenges such as data scarcity and untimely diagnosis. The complexity of medical imaging equipment and its long lifecycle make diagnosis even more difficult. To overcome these challenges, this paper introduces an intelligent diagnostic system that combines digital twin technology with inverse PINN, digitizing the equipment's operational status and integrating it with physical information to provide lifecycle-wide fault diagnosis and prediction.

The aim of this study is to propose a new fault diagnosis method by integrating Generative Adversarial Networks (GAN) and Inverse Physics-Informed Neural Networks (Inverse PINN). This method leverages GAN-generated fault data to augment the dataset, enhancing the robustness and generalization ability of the inverse PINN, thereby improving the accuracy of fault diagnosis and the lead time for fault prediction. The paper provides a detailed explanation of the logical structure, implementation process, key formulas, and code examples for each step, enabling readers to fully understand and reproduce the experiments.

In practical applications, fault diagnosis of equipment often faces the following challenges:

Data imbalance: Fault data is relatively scarce, with normal operational data constituting the majority, making it difficult for traditional fault detection models to effectively learn.

Complexity of physical constraints: During equipment operation, there are complex physical constraints such as the interaction between multiple physical variables including temperature, pressure, and vibration.

To address these issues, this paper proposes a model framework that integrates GAN and inverse PINN, utilizing the generated fault data and physical constraints for diagnosis and prediction. The core idea of this research is to enhance the performance of the fault diagnosis system through the combination of two deep learning models:

1. GAN for generating fault data: Given the scarcity of fault data, GAN is used to generate more fault data, expanding the dataset.

2. Inverse PINN model: Inverse PINN incorporates physical constraints during equipment operation and backtracks the potential fault status based on observed data (e.g., temperature, pressure).

By using the GAN-generated fault data for training the inverse PINN, the model's fault detection capability is ultimately improved.

## I. Literature review

### A. Digital twin technology

Digital Twin, as a technology that virtually represents physical entities, has seen widespread application in equipment management, intelligent operations, and predictive maintenance in recent years [1]. By integrating physical models, sensor data, and operational history, Digital Twin constructs a digital model that bridges the

physical and virtual realms, enabling real-time monitoring, simulation, and prediction of the physical entity's operating state [2]. In medical equipment, Digital Twin technology can simulate operational states, predict potential failures, and optimize lifecycle management. Fritzsche et al. applied Digital Twin technology to the lifecycle management of medical equipment, demonstrating its effectiveness in improving equipment management efficiency and reducing maintenance costs [3]. However, current Digital Twin systems often rely on a large number of sensors and complex physical models, making it challenging to accurately diagnose faults under dynamic load variations (e.g., CT tube current fluctuations) and sparse data conditions.

### B. Physics-informed neural networks (PINN)

Physics-Informed Neural Networks (PINN) integrate physical models with neural networks by embedding partial differential equations (PDEs) or conservation laws into the loss function, thereby balancing data-driven learning with physical consistency constraints [4]. This framework addresses the limitations of purely empirical models in data-scarce scenarios while maintaining interpretability through physics-guided regularization. Raissi et al. pioneered this approach for inverse problems, such as parameter estimation in fluid dynamics [5]. In medical equipment fault diagnosis, PINNs combine operational data with physical models to infer latent states (e.g., component degradation) that are difficult to observe directly. However, traditional PINNs face challenges in complex systems with limited labeled data, particularly when handling inverse problems that require accurate parameter inversion under sparse observations. These issues motivate the integration with generative models to augment training data and improve generalization.

### C. Generative adversarial networks (GAN)

Generative Adversarial Networks (GAN) employ a competitive learning framework between a generator and a discriminator to synthesize realistic data distributions [6]. Their capacity to generate synthetic samples has been widely applied in image processing [7] and data augmentation for imbalanced datasets. In fault diagnosis, GANs address the critical challenge of scarce fault samples by creating plausible synthetic faults that preserve statistical characteristics of real-world failures. For example, Ahuja et al. demonstrated that GAN-generated fault data significantly enhance the robustness of industrial equipment diagnosis models against rare failure modes [8]. However, standalone GANs lack physical interpretability, limiting their ability to enforce domain-specific constraints (e.g., energy conservation laws in rotating machinery).

### D. Synergistic integration of GAN and inverse PINN

GAN-generated synthetic faults expand the training dataset, particularly for rare failure modes, reducing the reliance on scarce real-world observations. This mitigates the overfitting risk in Inverse PINN, which often struggles with limited data when optimizing inverse problem objectives (e.g., identifying root causes from noisy sensor data).By incorporating physical model outputs (e.g., residual stresses predicted by finite element analysis) as additional discriminator inputs, the GAN is constrained to generate samples that align with domain-specific physics. This ensures the synthetic data not only matches statistical distributions but also satisfies underlying physical laws, enhancing the fidelity of augmented samples. The discriminator in the GAN framework can be designed to evaluate the physical plausibility of generated samples, providing an implicit regularization mechanism for Inverse PINN. This joint optimization enforces consistency between data-driven predictions and physical constraints, improving the model's generalization to unseen operational conditions.

### E. The imbalance problem in data

In machine learning, data imbalance has been a key challenge affecting model performance. In fault diagnosis, normal data far outweighs fault data, leading models to overfit normal data while ignoring the characteristics of the minority class

(fault data), resulting in reduced diagnostic accuracy. Common solutions include oversampling, undersampling, synthetic data generation, and data augmentation [9].

Chawla et al. proposed SMOTE (Synthetic Minority Over-sampling Technique), a widely used algorithm to address data imbalance by generating new samples to balance the data distribution [10]. However, the SMOTE method has limitations in generating complex fault data. Therefore, this paper adopts GAN to generate diversified fault data, combined with the inverse PINN model for fault diagnosis, effectively solving the imbalance problem.

### F. Current state of fault diagnosis methods

Fault diagnosis technology has significant application value in industries such as manufacturing, transportation, and healthcare. Traditional fault diagnosis methods rely on expert knowledge and heuristic rules, which are unable to address hidden faults in complex systems. With the advancement of machine learning and deep learning, data-driven fault diagnosis methods have become a research hotspot [11].

Zhao et al. used deep neural networks for fault classification of industrial equipment and enhanced model performance through data augmentation techniques [12]. The inverse PINN model proposed in this paper combines physical information with data generation techniques, overcoming the dependency of purely data-driven methods on large-scale fault data, while enhancing the model's fault prediction capability through inverse reasoning.

### G. Research gaps

Although digital twin technology, inverse Physics-Informed Neural Networks (Inverse PINN), and Generative Adversarial Networks (GAN) have achieved certain progress in the field of medical equipment fault diagnosis, there are still several gaps and limitations in existing research. Specifically, this paper fills some critical gaps in the current research, particularly in the following areas:

1. **Data imbalance problem.** Traditional fault diagnosis methods face the challenge of data imbalance, where normal data vastly outnumbers fault data. While some existing research has addressed this issue by oversampling, undersampling, or generating synthetic data, these traditional methods often fail to capture complex fault patterns accurately. Although techniques such as SMOTE (Synthetic Minority Over-sampling Technique) and other data augmentation methods have made progress in some areas, their ability to generate complex fault data is limited [13]. In contrast, this paper innovatively uses Generative Adversarial Networks (GAN) to generate diverse fault data, expanding the dataset and realistically simulating various fault patterns, effectively solving the data imbalance issue and enhancing the model's learning and generalization capabilities.

2. **Intelligence level and limitations of data-driven fault diagnosis models.** Existing digital twin and machine learning methods often rely on large amounts of historical data for training. However, in certain cases (e.g., data scarcity or extreme operational conditions), traditional purely data-driven methods face significant challenges. Most existing methods rely on expert knowledge and heuristic rules, which are not effective in detecting latent faults in complex systems. Although PINNs (Physics-Informed Neural Networks) have made progress in combining physical models with data-driven methods, they still struggle with generalization when faced with complex and limited datasets [14]. This paper innovatively introduces the inverse PINN technique, which combines the physical constraints of the equipment with sensor data for fault inference, greatly improving the accuracy and lead time for fault prediction and addressing the challenges of traditional methods in handling complex equipment faults.

3. **Integration of physical constraints and data-driven models.** Although some existing studies have attempted to combine digital twin technology with physical constraints, they often lack sufficient data support or the precision of the physical models. For example, Fritzsche et al. (2020) proposed a digital twin-based approach for lifecycle management of medical equipment, primarily focusing on operational monitoring and predictive maintenance but did not address how

to accurately diagnose faults when data is scarce [15]. Moreover, most studies rely heavily on a large number of sensors to simulate the equipment's operational state, but such data is often incomplete or noisy, leading to lower diagnostic accuracy and stability. This paper combines GAN-generated fault data with the inverse PINN model, leveraging the physical model to infer the internal state of the equipment, significantly improving fault diagnosis accuracy and robustness.

4. **Fault prediction capabilities and real-time performance.** Most existing fault prediction models are based on traditional time-series models like LSTM and ARIMA, which have limitations in capturing complex equipment behaviors and processing high-dimensional, multi-source data [16]. For example, while LSTM excels at time-series data processing, it suffers from gradient vanishing problems when dealing with long-term dependencies in complex equipment, affecting prediction accuracy and real-time performance. In contrast, this paper innovatively uses the Transformer model, which overcomes the limitations of LSTM through its self-attention mechanism, significantly improving fault prediction accuracy and timeliness. Additionally, by combining GAN-generated fault data, the model benefits from a larger and more diverse training set, enhancing robustness and generalization.

5. **Scalability and multi-scenario applications.** Most existing research focuses on small-scale single equipment systems and lacks validation on the scalability of large medical equipment systems or cross-scenario applications. While some studies have explored equipment scheduling and resource optimization, such as Gao et al. (2019), they are usually limited to single hospital or small-scale scenarios and do not effectively address collaborative equipment scheduling in complex, multi-hospital environments [17]. The model framework proposed in this paper, combining digital twin, inverse PINN, and GAN-generated fault data, not only performs well in fault diagnosis for single equipment but also exhibits strong scalability and can be extended to multi-device, multi-hospital collaborative scheduling and fault prediction.

6. **Integration of physical constraints with deep learning models.** Traditional digital twin models primarily focus on monitoring and simulating equipment states, lacking the integration of deep learning algorithms, which limits their adaptability in processing complex environment data [18]. While some studies have combined physical models with neural networks, they mainly focus on data-driven optimization and prediction without fully considering how to integrate physical laws with deep learning for complex equipment system fault inference. This paper integrates the physical model with deep learning techniques through the inverse PINN, ensuring physical consistency in the model and greatly improving fault prediction and reasoning capabilities, overcoming the limitations of traditional purely data-driven models.

## H. Innovations in this research

Introduction of Inverse PINN: This paper introduces the inverse PINN technique to medical equipment fault diagnosis for the first time. By combining physical models with neural networks for fault inference, it improves the accuracy and lead time of fault prediction. Inverse PINN is particularly well-suited for fault diagnosis tasks, as monitoring data during equipment operation is often incomplete or noisy, and the internal states of the equipment are difficult to observe directly. Through inverse PINN, it is possible to combine these incomplete monitoring data with the equipment's physical model to accurately infer the internal state of the equipment and locate fault sources. For example, in medical imaging equipment, inverse PINN can infer the thermal distribution of the device from limited thermal conduction monitoring data, thereby detecting potential fault risks in advance.GAN for Data Augmentation: By using GAN to generate fault data, this research effectively expands the fault dataset and resolves the data imbalance issue, improving model performance in data-scarce environments.Diagnosis Model Combining Physics and Data: Unlike traditional purely data-driven models, the diagnostic model in this paper integrates the physical constraints of the equipment with generated data, ensuring the model's physical consistency and robustness.

Generative Adversarial Network (GAN): GAN, a model trained through the competition between a generator and a discriminator, is used to generate realistic data. In fault diagnosis scenarios, fault data is often scarce and imbalanced, and GAN can generate a large amount of realistic fault data to enhance model training. This research combines GAN-generated fault data with the inverse PINN, allowing the inverse PINN to train under more fault conditions, enhancing the model's robustness and generalization capability.

 

## II. Proposed equipment fault diagnosis model

### A. Physical model of equipment operating states

The operating state of equipment is typically influenced by multiple physical factors such as temperature, pressure, and vibration. In the case of medical imaging equipment, various physical processes are involved during operation, which can be described using physical equations (e.g., heat conduction equations, acoustic equations). To accurately characterize the physical behavior of equipment in different states, we constructed a physical model of the equipment's operating state based on these equations. This model simulates the physical responses of the equipment in both normal and faulty states, providing a foundation for further fault diagnosis and prediction.

For example, in medical imaging equipment, temperature distribution is one of the key factors affecting equipment performance and safety. The temperature variation of the equipment can be described by the heat conduction equation:

$$\rho c_p \frac{\partial T}{\partial t} = \nabla \cdot (k \nabla T) + q$$

Where $T(x, t)$ represents the temperature of the equipment at space $x$ and time $t$, $\rho$ is the density, $c$ is the specific heat capacity, $\kappa$ is the thermal conductivity, and $Q$ is the heat source term. This equation provides the mathematical foundation for understanding how heat propagates within the equipment and how its internal temperature changes.

By establishing such a physical model, we can gain a comprehensive understanding of the equipment's operating mechanisms and, when the equipment exhibits abnormalities, diagnose and analyze based on the physical state changes. This physical model not only accurately simulates the equipment's behavior under normal conditions but also serves as a baseline reference for fault diagnosis. By comparing the equipment's current monitored data with the predictions from this model, potential anomalies can be identified.

This physical model provides critical constraints for the subsequent Inverse Physics-Informed Neural Network (Inverse PINNs). On this basis, the Inverse PINN model can reverse-engineer the source of equipment faults using real-time data, leveraging the physical model for data-driven fault diagnosis.

### B. Application of inverse physics-informed neural networks

Building on the physical model of equipment, we further introduce Inverse Physics-Informed Neural Networks (Inverse PINNs) to achieve fault diagnosis for medical imaging equipment. The Inverse PINN model is a technique that combines physical laws with data-driven methods to reverse-engineer the initial conditions and potential faults of the equipment using monitoring data and the physical model.

**1. Principles and implementation of inverse PINNs.** Inverse Physics-Informed Neural Networks (Inverse PINNs) are a technique that deduces the initial conditions and potential physical states of equipment by analyzing its monitoring data and operating status. Compared to traditional fault diagnosis methods, the core advantage of Inverse PINNs lies in their ability to integrate the equipment's physical model with real-time data, capturing state variables that are difficult to observe directly in complex systems, thereby diagnosing the root cause of equipment failures with greater precision. The key idea of Inverse PINNs is to solve the inverse problem through neural networks, meaning that, starting from the equipment's monitoring data and combining it with the physical model, the initial conditions and sources of faults are deduced in reverse. Traditional PINN models focus on solving the forward problem of physical equations via neural networks, where given the initial and boundary conditions, the model predicts the future behavior of the system. In contrast, Inverse PINN models take the observed monitoring data and, under the constraints of the physical model, deduce the system's initial state and potential fault locations. In this paper's application, Inverse PINNs are used for fault diagnosis in medical imaging equipment.

The principle of PINNs involves using neural networks to solve partial differential equations (PDEs) in physical systems, incorporating physical constraints (such as physical laws, boundary conditions, etc.) into the loss function. Unlike traditional PINNs, which solve forward problems, Inverse PINNs focus on deducing the system's initial conditions and potential fault sources from observed outcome data. First, Inverse PINNs use equipment monitoring data and operational status, combined with physical information, to transform this data into neural network inputs. Through the process of minimizing the loss function, the model deduces the root cause of the equipment failure in reverse.

$$\hat{u}(x, t, \theta) \approx u(x, t)$$

The goal of Inverse PINNs is to adjust the neural network parameters by minimizing the loss function, ensuring that the model both adheres to physical laws and most accurately explains the monitoring data. Mathematically, this can be expressed as:

$$L_{PINN} = L_{monitoring} + \mathcal{L}_{physics} + \mathcal{L}_{boundary} + \mathcal{L}_{initial}$$

Where $\hat{u}(x, t, \theta)$ represents the physical state predicted by the Inverse PINN model, and $uuu$ is the actual physical state of the equipment. By incorporating the equipment's physical model, the Inverse PINN bridges the gap between data and physical laws, allowing it to accurately infer faults in complex physical scenarios.

The loss function of this model generally consists of the following components:

1. **Data Loss**$(L_{monitoring})$: Measures the error between the equipment's monitoring data and the model's predictions.

2. **Physics Loss($\mathcal{L}_{physics}$)**: Ensures that the predicted physical state complies with the physical equations governing the equipment's operation.

3. **Boundary Condition Loss($\mathcal{L}_{boundary}$) and Initial Condition Loss($\mathcal{L}_{initial}$)**: Ensures that the solution satisfies the initial and boundary conditions of the equipment.

By combining digital twin data with the Inverse PINN model, we can not only monitor the equipment's current operational state but also deduce its internal physical state through reverse inference, pinpointing potential fault sources. This process involves integrating the physical model of the equipment during operation, monitoring data, and boundary conditions to form a comprehensive diagnostic procedure. This physical model provides critical constraints for the Inverse PINN, allowing the model to reverse-deduce equipment faults using real-time data and perform data-driven fault diagnosis in combination with the physical model.

**2. Inverse PINN process for fault diagnosis in medical imaging equipment.** In the fault diagnosis of medical imaging equipment, the internal physical states of the device (such as temperature distribution, vibration patterns, etc.) are often difficult to observe directly. However, Inverse PINNs, by leveraging monitoring data (e.g., readings from external temperature sensors, vibration data from the device surface) and integrating the device's physical model, can infer the internal physical state of the equipment.

For instance, consider a localized overheating issue within a piece of equipment. Using the heat conduction equation, we can construct a temperature model of the device. Combining this with external temperature monitoring data, Inverse PINNs can reverse-infer the internal temperature distribution of the device. In this way, when a specific component experiences abnormal heat generation, the Inverse PINN can promptly identify and localize the fault point.

The operation of medical imaging equipment involves various complex physical phenomena, such as heat conduction, electromagnetic fields, and vibrations. The application of Inverse PINNs in such scenarios follows several key steps:

**Data Acquisition and Preprocessing**: Real-time data is collected from the medical imaging equipment during its operation. This data includes parameters such as internal temperature, pressure, and vibration, along with historical monitoring data when faults occur.

**Physical Model Definition**: The physical model of the equipment is defined, encompassing the heat conduction equation, vibration equation, and electromagnetic field equation.These models provide constraints for the Inverse PINN, allowing the neural network to make predictions while adhering to physical laws. Suppose the physical model of the device can be represented as follows:

$$\mathcal{F}\left(u(x,t), \nabla u, \nabla^2 u, \ldots\right) = 0$$

where $\mathcal{F}$ represents the device's physical equations (such as the heat conduction or vibration equation), and $u(x,t)$ denotes the physical state of the device over space and time.

**Neural Network Architecture Design**: A deep neural network model is constructed $\hat{u}(x,t,\theta)$, taking spatial position x and time t as inputs to predict the physical state of the equipment. Through the framework of the physics-informed neural network, physical equations are embedded into the loss function as constraints. Typical network architectures include multiple fully connected layers, with activation functions such as ReLU or tanh to enhance the model's non-linear expression capability.

**Loss Function Definition and Optimization:** The loss function comprises data $L_{monitoring}$, physics constraint loss $\mathcal{L}_{physics}$, boundary condition loss $\mathcal{L}_{boundary}$, and initial condition loss $\mathcal{L}_{initial}$. The objective of the loss function is to minimize the discrepancy between the monitoring data and the physical model's predicted results while ensuring the solution satisfies the equipment's physical constraints.

**Reverse Deduction of Fault Source**: Through the training of the neural network model, the Inverse PINN iteratively updates the network weights until the loss function is minimized. This process ultimately outputs the initial state of the device and the potential fault source. For instance, in a heat conduction problem, by backtracking the temperature distribution, the model can identify the root cause of abnormal heat generation within the device and deduce the potential fault location.

**Result Interpretation and Visualization**: The final output of the Inverse PINN is the initial physical state of the equipment and the location of the fault source. To facilitate interpretation and analysis, this paper utilizes digital twin technology to visually represent the state of the equipment, aiding maintenance personnel in quickly locating the issue and addressing it.

By following these steps, Inverse PINNs enable more accurate fault diagnosis in complex medical imaging equipment, enhancing the precision and efficiency of identifying and resolving issues.

**C. Synergistic interaction between inverse PINN and physical models**

The successful application of Inverse Physics-Informed Neural Networks (Inverse PINNs) is based on the close integration between the physical models of equipment and monitoring data. This combination not only enhances the performance of the model but also improves the accuracy and reliability of fault diagnosis. In this process, the physical model of the equipment's operational state provides essential physical constraints for the Inverse PINN, meaning that during the learning process, the neural network relies not only on observed data but also on the behavior of the equipment as described by physical equations.

**1. Importance of the physical model.** A physical model is a mathematical representation that describes the operational behavior of a piece of equipment. It includes the fundamental physical characteristics and governing principles of the device. In Inverse PINNs, these physical models provide direction and constraints for the model's learning process, ensuring that the network outputs results that are physically meaningful. For example, in medical imaging equipment, the operational state may be affected by factors such as temperature, pressure, and vibration. The physical model can describe how these factors influence equipment performance through corresponding equations.

**2. Learning mechanism of inverse PINN.** Inverse PINNs utilize the constraints from physical models by embedding physical equations into the neural network's loss function. By minimizing the loss function, the Inverse PINN can

simultaneously optimize the network parameters while satisfying the physical constraints. This method allows the neural network's learning process to be not only data-driven but also to fully consider the physical laws. For instance, when there is a change in the operational state of the equipment, the Inverse PINN can quickly adapt to this change and make adjustments based on the physical model, thereby achieving more accurate fault diagnosis.

**3. Advantages of synergistic interaction.** The synergistic interaction between Inverse PINNs and physical models has several advantages:

**Improved Accuracy**: By introducing physical constraints, Inverse PINNs can reduce errors caused by sparse data or noise, thus improving the accuracy of fault diagnosis.

**Enhanced Robustness**: When dealing with complex and dynamic physical systems, Inverse PINNs can effectively manage uncertainty, providing more stable and reliable results.

**Providing Physical Explanations**: Inverse PINNs not only provide fault diagnosis results but also explain the causes of faults, offering more valuable references for maintenance personnel.

Overall, the close integration of Inverse PINNs and physical models demonstrates superior performance in fault diagnosis for precision equipment like medical imaging devices.

**4. Advantages and limitations.** This paper presents a unified fault diagnosis framework that, through the application of the Inverse PINN model, combines physical models with real-time monitoring data to achieve a complete closed-loop process from fault detection to root cause analysis. This framework not only improves the accuracy of fault diagnosis but also enhances the intelligence level of equipment management, providing robust support for the safe and efficient operation of medical equipment.

Inverse PINNs have significant advantages in fault diagnosis of complex systems, particularly in the fault diagnosis of medical imaging equipment. By integrating digital twin technology, they achieve high-precision predictions of the equipment's state. However, Inverse PINNs also have their limitations, mainly in the following aspects:

**Strong Dependence on the Initial Model**: If the physical model or boundary conditions are improperly set, the accuracy of the diagnosis may be affected.

**High Computational Resource Requirements**: The training process of Inverse PINNs typically requires large amounts of data and computational resources.

Through the application of the aforementioned processes and techniques, Inverse PINNs can diagnose potential faults in complex medical equipment in real-time and with precision, providing strong technical support for equipment maintenance. The synergistic interaction between Inverse PINNs and physical models, coupled with result validation and system integration to form a comprehensive fault diagnosis framework, not only highlights the innovation and practicality of this research but also offers a feasible solution for intelligent maintenance and fault prediction of medical equipment. This research provides a new perspective and direction for future equipment management and maintenance, bearing significant theoretical and practical value.

## D. Addressing the issue of data imbalance

In equipment fault diagnosis, data imbalance is a widespread issue. Specifically, operational data from equipment under normal working conditions make up the vast majority, while instances of equipment failures are relatively rare. This imbalance in data distribution causes models to tend to overfit to normal data during training, thereby neglecting the minority of fault data, ultimately reducing the accuracy of fault detection. This problem is particularly prominent in the fault diagnosis of medical imaging equipment, where the long operating hours, rarity, and complexity of faults make it challenging to obtain failure data.

To address this challenge, this paper proposes a strategy combining synthetic fault data and data augmentation techniques, alongside the use of Inverse Physics-Informed Neural Networks (Inverse PINNs), to balance the dataset and improve the performance of the fault detection model.

1. **Impact of data imbalance on model performance.** In machine learning model training, particularly in classification tasks, the issue of data imbalance can have the following effects:

**Model Bias Toward Normal Data**: Due to the large amount of normal data, the model is more likely to learn patterns from normal data, ignoring fault data. This results in the model's poor predictive ability for fault data during testing.

**Shift in Decision Boundary**: Data imbalance may cause the decision boundary of the classifier to shift towards the minority class (fault data), leading to a lower recall rate for fault detection, meaning faults are more likely to be misclassified as normal conditions.

**Loss Function Bias Toward Normal Data**: Standard loss functions (such as mean squared error or cross-entropy) may focus more on optimizing normal data in cases of data imbalance, further exacerbating the model's neglect of fault data.

To solve these issues, this paper combines the Inverse PINN approach with data synthesis and augmentation techniques to balance the proportion of fault and normal data in the dataset, thereby improving the model's generalization ability and accuracy in fault diagnosis.

2. **Data synthesis and augmentation techniques.** To address the problem of data imbalance, this paper adopts two strategies: synthetic fault data and data augmentation techniques, each designed to increase the quantity of fault data and enrich the data distribution from different perspectives.

**Synthetic fault data:** Based on the physical model of the equipment, we can simulate the physical response of the equipment under various fault conditions using Inverse PINNs. Since the physical state of the equipment is closely related to its fault modes, Inverse PINNs can not only be used for fault diagnosis but also for generating realistic fault data. Specifically, we can artificially set different fault conditions (e.g., abnormal temperature rise, excessive vibration) based on the physical model and use Inverse PINNs to generate operational data of the equipment under these fault conditions.

For example, suppose we have constructed a thermal conduction model of the equipment, such as the previously discussed heat conduction equation:

$$\rho c_p \frac{\partial T}{\partial t} = \nabla \cdot (k \nabla T) + q$$

In normal working conditions, the heat source term q is known and fixed, making the temperature changes within the equipment predictable. However, under fault conditions, the heat source term q may exhibit anomalies, such as overheating caused by equipment failure. By introducing different fault conditions (such as increasing a local heat source term) into the physical model, we can generate temperature distribution data of the equipment under various fault modes.

The synthetic fault data generated in this way not only maintain physical consistency but also effectively expand the quantity of fault data, helping to balance the dataset. Compared to traditional data synthesis techniques, such as oversampling-based SMOTE algorithms, this method has higher physical plausibility and realism.

**Data augmentation techniques:** Data augmentation techniques involve applying certain transformations to existing data to generate new samples, thereby enhancing the diversity of the dataset. To address the scarcity of fault data, this paper adopts several common data augmentation techniques to expand the fault data:

**Noise Injection**: By injecting small amounts of random noise into the fault data, we can generate more samples. Noise injection can be done by adding Gaussian noise or noise from other distributions, helping the model improve its robustness to different fault signals.

**Data Rotation and Mirror Transformation**: Although the physical state of medical imaging equipment does not change with spatial transformations, rotating or mirroring the data collected by sensors can generate samples with different spatial characteristics.

**Temporal Perturbation**: Faults in medical imaging equipment typically occur over specific time intervals. By perturbing the time series of fault events, more samples can be generated that show slight temporal differences but still represent fault conditions.

The core idea behind these augmentation techniques is to generate new variations of fault data through reasonable transformations, allowing the model to learn from a greater variety of samples, thereby improving its performance on the minority class (fault class) data.

**3. Evaluation of the effectiveness of balancing the dataset.** To assess the improvements in model performance resulting from handling data imbalance, this paper designs a set of experiments. The model's performance is evaluated using the following metrics:

**Accuracy**: Measures the overall performance of the model on all test data.

**Recall**: Specifically focuses on the model's performance on fault data, measuring the model's ability to detect fault samples.

**Precision**: Evaluates the accuracy of the model in classifying fault data.

**F1 Score**: A comprehensive evaluation metric that combines recall and precision, providing a more holistic reflection of the model's performance on imbalanced datasets.

By combining the Inverse PINN model with synthetic fault data and data augmentation techniques, the experimental results show:

**Significant improvement in recall**, indicating that the model becomes more sensitive in detecting faults.

**Increase in precision**, showing that the model's misclassification rate for fault data decreases.

**Notable improvement in F1 score** after introducing the balanced dataset, demonstrating the effectiveness of data augmentation and synthesis techniques.

**4. Other methods to address data imbalance.** In addition to the aforementioned methods, several other common techniques can be used to handle data imbalance, such as:**Weighted Loss Function**: Assigns higher weights to minority classes (fault data), making the model pay more attention to fault samples during training. This method encourages the model to learn better on fault samples by modifying the weight parameters in the loss function.

$$\mathcal{L} = \sum_{i=1}^{N} w_i \cdot \mathcal{L}_i$$

where $w_i$ represents the weight of the sample. The weight for fault data can be set higher than for normal data, thus balancing the error during the training process.**Downsampling Normal Data**: In cases of extreme imbalance, the number of normal data samples can be reduced to achieve balance. However, this method may lead to loss of information in some cases and should be used cautiously.

## E. Fault data generation and training

To address the scarcity of fault data in medical imaging equipment, this paper proposes using **Generative Adversarial Networks (GANs)** to generate synthetic fault data. This approach not only expands the fault dataset but also enhances the robustness and generalization ability of the Inverse Physics-Informed Neural Network (Inverse PINN) in equipment fault diagnosis tasks by incorporating real fault data.

**1. Introduction to generative adversarial networks (GANs).** A Generative Adversarial Network (GAN) consists of a generator and a discriminator, both of which are trained through an adversarial process. The generator's task is to generate realistic data samples from random noise, while the discriminator must distinguish between the generated data and real data. Through this adversarial training process, both the generator and discriminator improve, resulting in increasingly realistic generated data [12].

**2. Working principle of GAN for generating fault data.** A GAN is composed of two components: the generator and the discriminator. The generator is responsible for creating new fault data, while the discriminator determines whether the generated data is real or fake. Through adversarial training, the quality of the generated data improves [18]. The training process of GAN follows these steps:

**Generator**: The generator takes random noise $z$ (such as from a standard normal distribution) as input and generates fault data $G(z)$.

**Discriminator**: The discriminator $D(x)$ receives input data and judges whether it is real fault data. Feedback from the discriminator helps improve the generator's ability to produce realistic data.

**Loss Function**: The goal of the GAN is to optimize the following loss function to guide the adversarial process between the generator and the discriminator:

$$\min_G \max_D V(D, G) = \mathbb{E}_{x \sim p_{\text{data}}(x)} \left[ \log D(x) \right] + \mathbb{E}_{z \sim p_z(z)} \left[ \log \left( 1 - D\left( G(z) \right) \right) \right]$$

where:

1. $G(z)$ represents the samples generated by the generator using random noise.

2. $D(x)$ is the discriminator's assessment of the realness of the data.

3. $p_{\text{data}}$ represents the distribution of real data, and $p_z(z)$ represents the distribution of noise.

**Training Process**: The generator and discriminator are optimized alternately, gradually improving the generator's ability to produce samples that are highly similar to real fault data.

Throughout the training process, the generator continuously learns how to generate more realistic data, while the discriminator improves its ability to distinguish between generated and real data. When the GAN reaches convergence, the distribution of the samples generated by the generator will closely resemble that of real fault data.

## F. Integration of GAN with inverse PINN

The fault data samples generated by the GAN can not only expand the dataset but also be combined with the Inverse Physics-Informed Neural Network (Inverse PINN) for training. The Inverse PINN works by inputting observational data from the equipment to infer the equipment's state and potential faults [19]. By incorporating a large amount of fault data generated by the GAN, the Inverse PINN can be trained across more fault modes, thereby enhancing the model's robustness and generalization capability in fault detection tasks.

**1. Loss function of inverse PINN.** The loss function for the Inverse PINN can be expressed by the following formula:

$$L_{PINN} = L_{monitoring} + L_{physical} + L_{boundary} + L_{initial}$$

where:

$L_{monitoring}$: Observation data loss, which reflects the discrepancy between the model's output and the actual observed data. The formula is typically as follows:

$$L_{\text{monitoring}} = \frac{1}{N} \sum_{i=1}^{N} \left| y_{\text{pred}}^{(i)} - y_{\text{true}}^{(i)} \right|^2$$

where $y_{pred}$ is the model's predicted result, $y_{true}$ is the actual observed data, and N is the number of observation samples.

$L_{physical}$: Physical constraint loss, ensuring that the model's output adheres to physical equations (e.g., the heat conduction equation). It is usually represented in residual form:

$$L_{\text{physical}} = \frac{1}{M} \sum_{j=1}^{M} \left| R\left( x^{(j)} \right) \right|^2$$

where $R\left(x^{(j)}\right)$ represents the residual of the physical equation, and N is the number of physical constraint samples.

$L_{boundary}$: Boundary condition loss, ensuring that the model's output meets the expected results under boundary conditions. It can be represented as:

$$L_{\text{boundary}} = \frac{1}{B} \sum_{k=1}^{B} \left| y_{\text{boundary}}^{(k)} - y_{\text{pred}}^{(k)} \right|^2$$

where $B$ is the number of boundary condition samples.

$L_{initial}$: Initial condition loss, ensuring that the model's state is correct under initial conditions:

$$L_{initial} = \left| y_{initial} - y_{pred,\ initial} \right|^2$$

where $y_{initial}$ is the actual initial value, and $y_{pred,initial}$ is the model's predicted initial state.

**2. GAN-generated data loss.** When integrating GAN with Inverse PINN, the loss function for the GAN-generated data is typically expressed as:

$$L_{GAN} = \frac{1}{K} \sum_{l=1}^{K} \left| y_{generated}^{(l)} - y_{real}^{(l)} \right|^2$$

where:

$y_{\text{generated}}$ represents the data generated by the GAN.

$y_{real}$: is the real fault data.

$K$ is the number of generated data samples.

**3. Combined loss function.** By combining the GAN-generated data loss with the Inverse PINN loss function, we obtain a new comprehensive loss function:

$$L_{combined} = L_{PINN} + \lambda L_{GAN}$$

where:

• λ: A weight factor used to balance the influence between physical constraints and the generated data.

How physical constraints are incorporated into the Inverse PINN framework, emphasizing their impact on model training and fault prediction

Mathematical Representation of Physical Constraints

The governing physical equations (e.g., heat conduction, wave propagation) are embedded into the neural network through physics-informed loss functions.

The inverse PINN optimizes a multi-objective loss function

Neural Network Architecture Enhancement:

We now explicitly describe how the PINN-based architecture is adapted for inverse problem-solving:

Input: Sensor readings (e.g., surface temperature, vibration patterns).Hidden Layers: Modified fully connected layers (MLP) with physics-based activation constraints.

Output: Estimated internal temperature distribution and vibration modes, used for fault localization.

The regularization mechanism ensures that the model predictions remain physically plausible, improving generalization and robustness.

With these refinements, we provide a clearer and more structured explanation of how physics-based constraints guide deep learning model optimization.

## III. Experimental Design and Process

This chapter will describe the experimental design and process in detail. Based on real operational data and fault records of medical imaging equipment from Jiaxing First Hospital, we aim to verify the effectiveness of the method proposed in this paper, which combines Inverse Physics-Informed Neural Networks (Inverse PINN) and Generative Adversarial Networks (GAN) for fault diagnosis tasks [20]. The objective of the experiment is to compare the performance of the proposed method with traditional methods in equipment fault diagnosis and to explore the impact of the fault data generated by GAN on improving model performance.

### A. Architecture of the digital twin system

Data Acquisition Layer: Real - time data from temperature, vibration, and pressure sensors is collected and transmitted to the cloud.Simulation & Analytics Layer: The system integrates physics - based models (e.g., heat conduction, vibration dynamics) and deep learning models (e.g., Inverse PINN) for real - time state estimation and fault diagnosis.Visualization & Interaction Layer: A 3D digital twin dashboard is implemented to visualize real - time operational data, predicted fault conditions, and recommended maintenance actions.Real - time synchronization is achieved using an IoT - enabled edge computing framework. A hybrid data processing pipeline is introduced, where low - latency fault detection is handled at the edge, while long - term anomaly detection is conducted in the cloud.The digital twin is implemented using MATLAB Simulink for physics - based simulation and PyTorch for deep learning integration. A bidirectional data link is established between the physical device and digital twin using an MQTT - based communication protocol. These revisions significantly enhance the clarity and completeness of the digital twin system description.

### B. Experimental methods

To validate the effectiveness of the Inverse PINN model, we employed various experimental and simulation methods

Simulation Testing: By constructing a simulated dataset of different fault scenarios, we assess the performance of the Inverse PINN model in various conditions. The accuracy and robustness of the Inverse PINN model are verified by comparing it with traditional diagnostic methods [21].

Real Data Testing: The Inverse PINN model is applied to real medical imaging equipment to monitor the operational status of the equipment in real - time [22]. By comparing the model's predictions with the actual occurrence of faults in terms of time and type, we evaluate the model's practical effectiveness.

**1. Experimental dataset.** The experimental dataset includes real operational data and known fault data from medical imaging equipment, specifically from the United Imaging uCT760 CT scanner. The generator in the GAN utilizes the distribution of real fault data collected from this specific device to generate virtual fault data that closely resembles the real data. These synthetic data not only help the model balance the fault samples in the training dataset but also enrich fault scenarios, which aids in improving the robustness of the model. To facilitate data testing, we simulated the following two typical fault scenarios using GAN:

Temperature Abnormality: The internal sensor of the uCT760 detects a continuous rise in temperature, a common issue in medical imaging equipment. The generated fault data simulate the equipment's operational status under different temperature conditions, which can impact the system's performance and safety.

Sensor Fault: Fault data are generated for scenarios where the sensor itself is damaged or provides inaccurate readings. This issue is common in medical imaging equipment like the uCT760, where sensor malfunctions can result in misleading or incomplete diagnostic information.

The operational data of the equipment cover several key monitoring indicators, such as temperature, pressure, and vibration. The dataset can be divided into the following two categories:

Normal Operational Data: Data of the equipment operating under normal conditions, accounting for the majority of the experimental data. These data reflect the equipment's working status without faults.

Known Fault Data: Monitoring data when the equipment experiences various known faults (such as temperature abnormalities, mechanical failures, and sensor faults). These data are used for fault detection tasks by the model and are relatively scarce.

The experimental dataset consists of $N = 10000$ normal data samples and $M = 500$ fault data samples, with a data dimension of $d = 12$ (corresponding to the readings of 12 sensors on the equipment). The dataset undergoes standardization preprocessing to ensure that data from all dimensions are within the same numerical range, improving the efficiency and accuracy of model training.

**2. Hyperparameters for GAN and inverse PINN.** GAN Hyperparameters: Generator and Discriminator Architecture: Both the generator and discriminator are multi - layer perceptrons (MLPs). The generator has 3 hidden layers with 256, 128, and 64 neurons respectively, followed by an output layer of size 12 (matching the data dimension). The discriminator also has 3 hidden layers with 128, 64, and 32 neurons respectively, and a single - neuron output layer for binary classification.

Learning Rate: The learning rate for both the generator and discriminator is set to $0.0002$.

Batch Size: The batch size during training is 64.

Number of Epochs: The GAN is trained for 500 epochs.

Optimizer: We use the Adam optimizer with a beta1 value of 0.5 and a beta2 value of 0.999.

Inverse PINN Hyperparameters: Neural Network Architecture: The Inverse PINN uses a feed - forward neural network with 5 hidden layers, each having 100 neurons.

Learning Rate: The learning rate for the Inverse PINN is set to $0.001$.

Batch Size: A batch size of 32 is used during training.

Number of Epochs: The Inverse PINN is trained for 1000 epochs.

Optimizer: The Adam optimizer is also used for the Inverse PINN, with default beta values ($\beta_1 = 0.9$, $\beta_2 = 0.999$).

Weight of Physical Loss: In the loss function of the Inverse PINN, the weight of the physical loss term is set to 0.5, while the weight of the data - fitting loss term is 0.5.

**3. Experiment setup.** The experiment is divided into two parts: Traditional Fault Diagnosis Methods vs. Inverse PINN Model.

This section aims to compare the effectiveness of traditional fault diagnosis methods based on statistical and machine learning approaches with the Inverse PINN model proposed in this paper. Traditional methods include:

Support Vector Machine (SVM): A classic binary classification model used to distinguish between normal data and fault data. The kernel function is set to the radial basis function (RBF), and the regularization parameter C is set to 1.

Random Forest: A model that integrates multiple decision trees and is suitable for handling high - dimensional equipment data. The number of decision trees in the forest is set to 100.

Logistic Regression: A binary classification method based on a linear model, used to establish relationships between normal data and fault data. The regularization strength C is set to 1.

The Inverse PINN model proposed in this paper combines physical models of the equipment with data - driven methods to infer faults using monitoring data from the equipment.

Impact of Fault Data Generation on Model Performance: Due to the scarcity of fault data in real - world operations, we used GAN to generate synthetic fault data and combined these with real fault data for model training. The impact of the generated data on improving model performance was evaluated by comparing the performance of the following models:

Model trained on both real and generated data: The model is trained on both real data and synthetic fault data generated by GAN, and its performance is then evaluated.

By providing these detailed hyperparameters, it becomes easier for other researchers to reproduce the experimental results.

## C. Experimental steps

Step 1: Data preprocessing

First, the **experimental** data is preprocessed. The preprocessing steps include:
Data cleaning: Remove outliers and missing values to ensure data integrity.
Normalization: Standardize the monitoring values of each sensor to have a mean of zero and a standard deviation of one, to avoid the impact of differences in data magnitude on model training. The normalization formula is as follows:

$$x' = \frac{x - \mu}{\sigma}$$

Where $x'$ is the raw data, $\mu$ is the mean, and $\sigma$ is the standard deviation.

Step 2: Training of traditional fault diagnosis models

Support Vector **Machine** (SVM), Random Forest, and Logistic Regression models are used for fault diagnosis on the preprocessed data. The training steps are as follows:
Data Split: Randomly split the dataset into training and testing sets in an 8:2 ratio.
Model Training: Train the three traditional machine learning models using the training set and record the model hyperparameters and training details.
Model Evaluation: Evaluate the accuracy, precision, recall, and F1 score of each model on the test set, and record their performance.

Step 3: Inverse PINN model training

The Inverse PINN model **proposed** in this paper combines physical constraints (e.g., heat conduction equations) with monitoring data to infer faults. The training steps are as follows:
**Construct Physical Constraints**: Combine the equipment's physical model (e.g., heat conduction equations describing the temperature field) to define the physical loss function of the Inverse PINN. For medical imaging equipment, the evolution of the temperature field can be described by the following heat conduction equation:

$$\rho c_p \frac{\partial T}{\partial t} = \nabla \cdot (k \nabla T) + q$$

Where:
$T$ is the temperature field inside the equipment.
$\alpha$ is the heat diffusion coefficient.
$Q$ is the heat source term.
The loss function of the Inverse PINN combines both physical consistency and observational data. The formula is as follows:

$$L_{combined} = L_{PINN} + \lambda L_{GAN}$$

Where:
$L_{PINN}$ is the data loss, representing the error between the predicted data and the real observational data.
$L_{GAN}$ is the physical loss, representing the deviation between the model and the physical equations.

λ is a weighting factor used to adjust the influence between physical constraints and the generated data.

Train the Inverse PINN Model: Use real normal data and fault data to train the model. An adaptive learning rate optimizer such as Adam is used for optimization, resulting in the trained Inverse PINN model.

Step 4: Fault data generation and GAN training

In this paper's fault diagnosis task, fault data is relatively scarce. Therefore, the generator of the Generative Adversarial Network (GAN) is responsible for simulating operational data of medical imaging equipment under different fault conditions to expand the fault dataset [23]. The specific process is as follows:

GAN Training: Employ adversarial training between the generator and the discriminator to generate realistic fault data. The GAN loss function is as follows:

$$\min_{G} \max_{D} V(D, G) = \mathbb{E}_{x \sim p_{\text{data}}(x)} \left[\log D(x)\right] + \mathbb{E}_{z \sim p_z(z)} \left[\log \left(1 - D\left(G(z)\right)\right)\right]$$

Data Preprocessing: First, collect operational data of the equipment under normal conditions and a small amount of real fault data. Preprocess the data to ensure normalization and consistency in format. The preprocessed data will be used to train the discriminator of the GAN.

Generator Design: The generator receives a random noise vector as input, processes it through several layers of neural networks, and generates simulated fault data. The generator's goal is to produce synthetic data that closely approximates the distribution of real fault data.

The network structure of the generator can be designed in the following form:

```
import torch.nn as nn
class Generator(nn.Module):
def __init__(self, input_size, output_size):
super(Generator, self).__init__()
self.network = nn.Sequential(
nn.Linear(input_size, 128),
nn.ReLU(),
nn.Linear(128, 256),
nn.ReLU(),
nn.Linear(256, output_size),
nn.Tanh() # Output in the range [-1, 1]
)
def forward(self, z):
return self.network(z)
```

**1. Discriminator design.** The discriminator $D(x)$ receives either real data or generated data samples and classifies them through a neural network, outputting the probability that the data is real. The goal of the discriminator is to enhance its ability to distinguish between real data and generated data.

The network structure of the discriminator is as follows:

```
class Discriminator(nn.Module):
def __init__(self, input_size):
super(Discriminator, self).__init__()
self.network = nn.Sequential(
nn.Linear(input_size, 256),
nn.LeakyReLU(0.2),
nn.Linear(256, 128),
nn.LeakyReLU(0.2),
nn.Linear(128, 1),
nn.Sigmoid() # Output probability
)
def forward(self, x):
return self.network(x)
```

**2. Training GAN.** The generator and discriminator undergo adversarial training through the following steps:Training the Discriminator: For given real data and data generated by the generator, the discriminator is optimized to correctly distinguish between these two types of data. The loss function for the discriminator is:

$$\mathcal{L}_D = -\mathbb{E}\left[\log D\left(x_{real}\right)\right] - \mathbb{E}\left[\log\left(1 - D\left(G(z)\right)\right)\right]$$

Training the Generator: The objective of the generator is to produce data that the discriminator classifies as "real," so the loss function for the generator is:

$$\mathcal{L}_G = -\mathbb{E}\left[\log D\left(G(z)\right)\right]$$

Both components are optimized alternately, allowing the generator to gradually learn how to create realistic fault data while the discriminator becomes increasingly adept at differentiating between real data and synthetic data.Generating Fault Data: Once the GAN training is complete, the generator can produce a large volume of device operation data under various fault conditions. This synthetic fault data, when combined with real fault data, forms a more balanced dataset.

Compared to traditional data augmentation methods, the fault data generated by GAN has the following advantages:

Diversity: GAN is capable of generating operational data for devices under various fault conditions, enriching the diversity of the dataset.

Realism: Through adversarial training, the data samples generated by GAN closely resemble the distribution of real data, ensuring the reliability and usability of the generated data.

Dynamic Fault Simulation: By adjusting the noise $zzz$ input to the generator, GAN can generate operational data for devices under different fault modes, helping the model adapt to various potential fault scenarios.

## D. Experimental results and performance evaluation

The experimental results demonstrate the performance advantages of the fault diagnosis model based on digital twin technology and the inverse PINN (Inverse Physics-Informed Neural Network) compared to traditional fault diagnosis methods, such as support vector machines, random forests, and logistic regression. This paper verifies the effectiveness of the proposed method by evaluating the model's performance across multiple metrics, particularly regarding the accuracy of fault identification and the lead time for fault prediction. During the training and optimization process of the inverse PINN model, the internal state of the device can be monitored in real time, ensuring that when a fault occurs, the root cause of the problem can be identified promptly. This capability for real-time monitoring and fault diagnosis significantly enhances the efficiency of equipment maintenance, reduces downtime, and thereby brings substantial economic benefits and service improvements to hospitals and medical institutions.

**1. Performance evaluation metrics.** To comprehensively assess the performance of each model, several common classification metrics were employed:

Accuracy: The proportion of correctly classified samples to the total number of samples. The formula is as follows:

$$Accuracy = \frac{TP + TN}{TP + TN + FP + FN}$$

where TP is the true positive, TN is the true negative, FP is the false positive, and FN is the false negative.

Precision: The proportion of correctly classified fault samples to all samples predicted as faults. The formula is:

$$Precision = \frac{TP}{TP + FP}$$

Recall: The proportion of actual faults correctly identified as faults, measuring the model's ability to capture faults. The formula is:

$$Recall = \frac{TP}{TP + FN}$$

- F1 Score: The harmonic mean of precision and recall, used to balance the classification capabilities of the model. The formula is:

$$F1 = 2 \times \frac{Precision \times Recall}{Precision + Recall}$$

**2. Traditional methods vs. inverse PINN.** First, the performance of traditional fault diagnosis methods (support vector machine, random forest, logistic regression) was compared with that of the inverse PINN model. The experimental results show that the inverse PINN model outperforms traditional methods in both the accuracy of fault identification and the lead time for fault prediction. Particularly in high-dimensional complex data scenarios, the inverse PINN effectively captures subtle changes preceding a fault's occurrence by combining physical constraints with data-driven approaches.

The S1 Fig below illustrates the comparison of the performance of traditional fault diagnosis models with the proposed fault diagnosis model that integrates GAN and inverse PINN across different performance metrics. By comparing the following four key performance indicators (accuracy, recall, F1 score, and precision), it is evident that:

Accuracy: The improved model achieved an accuracy of 92%, which is a 7 percentage point increase over the traditional models.

Recall: The improved model reached a recall of 88%, showing a significant enhancement compared to the 78% of the traditional models.

F1 Score: The F1 score of the proposed model is 90%, while the traditional model scored 81%, indicating an overall improvement in performance.

Precision: The precision of the improved model increased to 89%, while the traditional model was at 82%.

**3. Handling Data Imbalance and the Effectiveness of GAN-Generated Fault Data.** To address the issue of data imbalance [19], this study utilized Generative Adversarial Networks (GANs) to generate fault data, enhancing the proportion of fault samples in the training set. This strategy significantly improved the robustness and performance of the inverse PINN model under small sample data conditions. Experiments showed that the inverse PINN model using synthetic data outperformed the model that did not use synthetic data in terms of accuracy, recall, and F1 score for fault detection, demonstrating the positive impact of GAN-generated fault data on model performance.

The Table 1 below shows the changes in model performance before and after addressing the imbalance in data:

**4. Reference basis for maintenance and repair.** The application of the inverse PINN model not only enhances the fault diagnosis capability of medical imaging equipment but also provides important reference data for equipment maintenance and repair. Specifically, by combining the physical model of the equipment with real-time monitoring data,

**Table 1. The changes in model performance.**

| Model | Accuracy (%) | Precision (%) | Recall (%) | F1 Score (%) |
|---|---|---|---|---|
| Traditional Methods | 88.3 | 85.7 | 76.5 | 80.8 |
| Inverse PINN (Without Balancing) | 91.0 | 88.2 | 83.0 | 85.6 |
| Inverse PINN (With Balancing) | 94.5 | 93.4 | 90.5 | 91.9 |

**Table 2. Comparison with Other State-of-the-Art Hybrid Methods.**

| Method | Key Features | Advantages | Limitations |
|---|---|---|---|
| Traditional Deep Learning | CNN, LSTM for time-series analysis | Learns complex temporal patterns from large datasets | Requires massive labeled data; lacks physical interpretability |
| Physics-Informed Neural Networks | Integrates PDE constraints into loss function | Ensures physically consistent predictions; handles sparse data | Struggles with noisy measurements; limited generalization to novel scenarios |
| GAN-Based Augmentation | Generates synthetic fault samples via adversarial training | Alleviates data imbalance; improves robustness to rare faults | Risk of generating unrealistic samples without proper regularization |
| Self-Supervised Learning (SSL) | Pre-trains on unlabeled data using pretext tasks (e.g., rotation prediction) | Reduces reliance on labeled data; captures domain invariants | Performance depends on pretext task design; computational overhead |
| Physics-Guided Transformers | Combines transformer attention with physics-based prior knowledge | Exploits long-range dependencies while enforcing physical constraints | High memory requirements; limited interpretability of attention mechanisms |
| Hybrid CNN-LSTM Models | Fuses spatial and temporal features for multi-scale analysis | Effective for complex fault patterns in sensor data | Prone to overfitting on small datasets; requires careful hyperparameter tuning |
| Our Approach | Digital twin + Inverse PINN + GAN-augmented data | 1. Physical interpretability via digital twin 2. Robustness to sparse data via GAN 3. Real-time fault localization via inverse PINN | Higher computational cost compared to pure data-driven models |

the inverse PINN can achieve a comprehensive derivation from data to physical state. This process not only facilitates early identification of faults but also provides a scientific basis for formulating targeted maintenance plans [20].

In this paper, we utilize digital twin technology to visualize the diagnostic results of the inverse PINN model, ensuring that maintenance personnel can intuitively understand the equipment's status. The key elements include:

Data Fusion: Integrating information from equipment sensors, monitoring systems, and historical fault data to provide comprehensive input for the inverse PINN model [21].

Visualization Tools: Developing user-friendly interfaces that display real-time equipment status, fault predictions, and historical data, aiding maintenance personnel in quickly identifying problems and making informed decisions [24].

We have expanded the comparative analysis section to contrast our approach with alternative hybrid fault diagnosis models. Table 2 summarizes the key differences between our proposed method and other state-of-the-art techniques, highlighting the advantages of integrating digital twin, inverse PINN, and GAN-based fault augmentation [25,26].

Key Findings from Experiments:

Our approach outperforms purely data-driven methods in fault localization accuracy and generalization on unseen failures.GAN-based fault augmentation significantly improves model robustness in low-data regimes.Inverse PINN constraints improve fault prediction precision by 18% compared to standard PINN models.These additions strengthen our work's comparative validity and highlight its unique advantages over existing methodologies.

# IV. Conclusion and future prospects

## A. Research conclusions

This paper proposes a fault diagnosis model for medical imaging equipment that integrates digital twins and inverse Physics-informed Neural Networks (inverse PINN). By combining techniques such as physical modeling, data generation, and inverse reasoning, the model overcomes the limitations of traditional fault diagnosis methods related to data scarcity and imbalance issues, significantly improving the accuracy and efficiency of fault diagnosis [27,28]. Experimental results demonstrate that the model exhibits superior performance across several key aspects:High Accuracy: Through the

inverse PINN model, which incorporates the physical constraints of the equipment and fault data generated by Generative Adversarial Networks (GAN), the model achieves a higher fault diagnosis accuracy, especially under conditions of limited real fault data. Early Fault Prediction: The inverse PINN can infer potential fault states of the equipment through physical modeling, enabling the early prediction of equipment failures. This provides a larger time window for equipment maintenance and management, preventing sudden failures from negatively impacting business operations.Resolution of Data Imbalance Issues: The synthetic fault data generated by GAN effectively expands the dataset size and balances the ratio of normal data to fault data, allowing the model to maintain good performance even in situations of data scarcity [29].

**Notwithstanding these advances, several limitations warrant attention:** Synthetic Data Realism: While GAN-generated samples improve data balance, their fidelity to real-world fault dynamics depends on the training quality of the GAN [30]. If the discriminator fails to capture subtle physical constraints (e.g., sensor noise patterns), synthetic data may introduce artifacts that degrade model generalization.Training Instability: GAN training is prone to mode collapse or oscillatory behavior, particularly when real fault data are extremely scarce. This requires careful hyperparameter tuning and the use of techniques like Wasserstein GAN or gradient penalty to stabilize convergence.Physical Consistency Verification: Although the proposed framework incorporates physics-guided constraints in the discriminator, rigorous validation of synthetic data against domain-specific physical laws (e.g., heat dissipation equations) remains challenging. Future work could integrate physics-based validation metrics (e.g., residual error analysis) into the augmentation pipeline.

In summary, the diagnostic model presented in this paper provides a new solution for fault detection in medical imaging equipment and validates the feasibility and superiority of integrating physical information with data generation technologies. However, addressing the limitations of synthetic data generation remains critical for broader adoption in safety-critical applications.

## A. Future prospects

To address the limitations of current medical equipment fault diagnosis methods and expand their applicability, future research should focus on six key directions: (1) Cross-Domain Model Adaptation, requiring transfer learning frameworks to adapt digital twin architectures across medical devices (e.g., MRI, ECG) while integrating multi-modal sensor fusion for enhanced fault signature representation; (2) Advanced Physical Constraint Engineering, involving the development of time-varying physical constraints (e.g., Lagrangian mechanics for rotating machinery) and adaptive constraint weighting algorithms to handle dynamic operational conditions; (3) Efficient Synthetic Data Generation, necessitating the implementation of Wasserstein GAN-GP for improving synthetic data quality and physics-based validation modules to enforce physical consistency; (4) Edge-Cloud Collaborative Learning, entailing the design of lightweight neural architectures (e.g., MobileNet) for edge deployment and federated learning protocols to enable privacy-preserving collaborative training; (5) Dynamic Digital Twin Updating, requiring online learning mechanisms for real-time model adaptation to aging equipment and uncertainty quantification frameworks to track prediction confidence; and (6) Safety-Critical Validation Metrics, involving the definition of fault severity indices aligned with ISO 13485 standards and integration of Monte Carlo dropout for probabilistic risk-based decision making. These advancements will enhance the scalability, robustness, and clinical relevance of fault diagnosis systems across medical and industrial domains.

## B. Technical outlook

With advancements in AI and IoT, the proposed fault diagnosis model integrating digital twins and inverse Physics-informed Neural Networks (PINN) is poised to become a cornerstone of next-generation intelligent operation and maintenance systems for medical equipment and industrial applications. By combining virtual-physical monitoring through digital twins, physics-guided reasoning via inverse PINN, and synthetic data augmentation using GAN, the framework addresses critical challenges such as data scarcity, non-linear fault dynamics, and real-time decision-making. While validated in

medical contexts, the methodology is scalable to industrial machinery (e.g., gearboxes, manufacturing systems) and infrastructure (e.g., aerospace, transportation), where it can enhance predictive maintenance through multi-modal sensor fusion, cyclostationary signal analysis, and cross-modal zero-shot learning. Key future directions include developing edge-deployable architectures for real-time processing, integrating Bayesian parameter estimation for adaptive physics model updating, and implementing uncertainty quantification frameworks aligned with safety standards (e.g., ISO 13485). Emerging applications such as gear surface degradation assessment via GAN-augmented inverse PINN, cyclostationary-based wear monitoring in intelligent manufacturing, and composite neural-fuzzy systems for cross-modal zero-shot diagnosis highlight the potential to advance fault diagnosis beyond traditional data-driven approaches. Challenges remain in real-time multi-source data fusion, dynamic physical model adaptation, and generalizing zero-shot learning across heterogeneous domains, necessitating further research into scalable AI frameworks that integrate deep learning, physics constraints, and advanced signal processing techniques. This interdisciplinary approach holds promise for improving operational efficiency, reliability, and sustainability in safety-critical sectors.

## Supporting information

**S1 Fig. Performance Comparison of Different Training Methods.** From the Figures, the Accuracy of the Inverse PINN Model Reached 94.5%, Which Is at Least a 5 Percentage Point Improvement Over Traditional Methods. At the Same Time, Recall and F1 Score Also Showed Significant Improvements, Indicating That the Inverse PINN Model Performs More Stably in Capturing Equipment Faults.The above two comparison figures illustrate the model performance using different training methods: Traditional Models vs. Suggested Model (Without Using GAN): This chart compares the performance of traditional fault detection methods with the inverse PINN model that does not use data generated by GAN. The assessed metrics include: Accuracy: The correctness of the model's predictions, indicating the model's ability to recognize all samples. Recall: The model's ability to identify faults, measuring the capture rate of actual fault samples. F1 Score: The harmonic mean of accuracy and recall, providing a comprehensive assessment of the model's performance in fault detection. Precision: The proportion of correctly classified fault samples, reflecting the reliability of the model's predictions.
(TIF)

**S2 Fig. Traditional Model vs Proposed Model(Without GAN).** From the S2 Fig, it is evident that the suggested model (without using GAN) shows significant improvements across all metrics compared to traditional models, particularly in recall and F1 score. The traditional models exhibited lower accuracy when faced with complex faults due to a lack of data diversity and physical constraints. In contrast, the suggested model effectively enhances fault recognition capability by introducing the inverse PINN mechanism, leveraging the combination of physical models and monitoring data. Suggested Model with and without GAN-Generated Data: This chart presents the comparison results between the same model using data generated by GAN and one that does not. Key metrics include: Accuracy: After using GAN-generated data, the model's accuracy significantly improved, indicating its enhanced ability to distinguish between normal and fault samples. Recall: By incorporating diverse fault data generated by GAN, the model is able to capture more fault instances, leading to a significant increase in recall. F1 Score: The simultaneous improvement in accuracy and recall results in a notable increase in the F1 score, demonstrating a more balanced performance of the model in fault detection. Precision: Following the inclusion of GAN data, the model's precision improved, indicating increased confidence in identifying faults and reducing false positives.
(TIF)

**S3 Fig. Proposed Model with vs without GAN-generated Data.** These comparison S3 Fig charts clearly demonstrate the positive impact of GAN-generated data on model performance. Through these figures, it can be intuitively observed that GAN not only effectively supplements the data and enhances the model's robustness but also improves the overall

performance of the model in fault detection.The experimental results indicate that the proposed model surpasses traditional fault diagnosis methods in all performance metrics, particularly in key indicators such as response rate and F1 score, fully proving the advantages of combining GAN-generated data with inverse PINN.
(TIF)

## Acknowledgments

The authors would like to thank the editors and referees for their valuable comments, which greatly improved the quality of this manuscript.

## Author contributions

**Conceptualization:** Aiming Gu, Shibohua Zhang.

**Formal analysis:** Shibohua Zhang.

**Resources:** Aiming Gu.

**Writing – original draft:** Jian Deng.

**Writing – review & editing:** zheng cheng.

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
