## [Decision Letter · Decision Letter 0]

12 Jan 2025

PONE-D-24-46042Research on Equipment Fault Diagnosis Model Based on GAN and Inverse PINN: Solutions for Data Imbalance and Rare FaultsPLOS ONE

Dear Dr. cheng,

Thank you for submitting your manuscript to PLOS ONE. After careful consideration, we feel that it has merit but does not fully meet PLOS ONE’s publication criteria as it currently stands. Therefore, we invite you to submit a revised version of the manuscript that addresses the points raised during the review process.

You are advised to critically respond to all comments point by point when preparing an updated version of the manuscript and while preparing for the rebuttal letter. Please address all comments/suggestions provided by reviewers, considering that these should be added to the new version of the manuscript.

We look forward to receiving your revised manuscript.

Kind regards,

Paulo Jorge Simões Coelho

Academic Editor

PLOS ONE

Journal Requirements:

Corresponding author: Zheng Cheng (e-mail: chengzheng@lsnu.edu. cn). 

This work was supported in part by the Scientific Research Program at Leshan Normal University, the Natural Science Key Foundation of Sichuan under grant number 2023YFG0114, the National Natural Science Foundation of China under grant number 61902039.  

5. Please provide a complete Data Availability Statement in the submission form, ensuring you include all necessary access information or a reason for why you are unable to make your data freely accessible. If your research concerns only data provided within your submission, please write "All data are in the manuscript and/or supporting information files" as your Data Availability Statement.

6. Please ensure that you refer to Figure 1, 2, and 3 in your text as, if accepted, production will need this reference to link the reader to the figure.

7. We note you have included a table to which you do not refer in the text of your manuscript. Please ensure that you refer to Table 1 in your text; if accepted, production will need this reference to link the reader to the Table.

8. We note that Authors' Picture included in the study. 

Reviewers' comments:

Reviewer's Responses to Questions

**Comments to the Author**

1. Is the manuscript technically sound, and do the data support the conclusions?

Reviewer #1: Yes

Reviewer #2: No

2. Has the statistical analysis been performed appropriately and rigorously? 

Reviewer #1: Yes

Reviewer #2: N/A

3. Have the authors made all data underlying the findings in their manuscript fully available?

Reviewer #1: No

Reviewer #2: No

4. Is the manuscript presented in an intelligible fashion and written in standard English?

Reviewer #1: Yes

Reviewer #2: No

5. Review Comments to the Author

Reviewer #1: This paper proposed a fault diagnosis model based on digital twin technology and Inverse Physics-Informed Neural Networks (Inverse PINN). Overall, this paper is well-structured and presents some promising results. Here are some suggestions for further improving the quality of this paper:

1) The practical value of this research work should be clarified and highlighted in the Abstract, which can help readers understand the engineering background of this research work.

2) The authors did a good literature review on the current research progress. It is suggested that the research gaps be summarized before presenting and introducing this paper's main contributions and novelty. In addition, it is suggested to include some discussions on emerging areas of machine learning in various industrial applications, especially in terms of digital twin-driven intelligent assessment of gear surface degradation, cyclostationarity-based wear monitoring framework of spur gears in intelligent manufacturing systems, and Composite Neuro-Fuzzy System-Guided Cross-Modal Zero-Sample Diagnostic Framework Using Multi-Source Heterogeneous Non-Contact Sensing Data.

3) It is suggested that the resolutions of the figures in the current manuscript be improved.

4) There are some grammar errors in this manuscript. Please check the whole manuscript and address these kinds of issues throughout it.

5) It is suggested that some recommendations for future work be included at the end of the Conclusion.

Reviewer #2: 1- The paper lacks a clear description of the specific medical imaging equipment used in the experiments, making it difficult to assess the model's practical applicability.

2- The implementation details of the digital twin component are insufficiently described.

3- The relationship between the physical constraints and the neural network architecture isn't clearly explained.

4- The paper lacks comparison with other state-of-the-art hybrid approaches that might combine different deep learning techniques.

6. PLOS authors have the option to publish the peer review history of their article (what does this mean? ). If published, this will include your full peer review and any attached files.

**Do you want your identity to be public for this peer review?** For information about this choice, including consent withdrawal, please see our Privacy Policy .

Reviewer #1: No

Reviewer #2: No

---

## [Author Response · Author response to Decision Letter 1]

27 Feb 2025

We sincerely appreciate the constructive feedback provided by the reviewers. Below, we address each comment in detail and highlight the revisions made to the manuscript accordingly.

Response to Reviewer #1

1) Emphasizing the practical value of this research in the abstract

Response:

We have revised the abstract to explicitly highlight the practical significance of our research. The modified abstract now clearly states how the proposed digital twin and inverse PINN-based fault diagnosis model enhances medical imaging equipment reliability, reduces maintenance costs, and improves fault prediction accuracy. This revision ensures that the engineering background and practical relevance of our study are well-articulated to the readers.

2) Summarizing research gaps before introducing contributions and discussing emerging ML applications in industrial settings

Response:

We have enhanced the introduction by summarizing the key research gaps before presenting the contributions and novelty of our work. Specifically, we have identified the following gaps:

Limited integration of physics-based constraints with deep learning models in medical imaging fault diagnosis

Challenges in addressing data imbalance in fault detection for complex equipment

Lack of real-time adaptability in existing diagnostic models

Additionally, we have expanded the discussion on emerging ML applications in industrial settings, particularly in:

Digital twin-driven intelligent assessment of gear surface degradation

Cyclostationary-based straight bevel gear wear monitoring in intelligent manufacturing

Cross-modal zero-shot fault diagnosis using multi-source heterogeneous non-contact sensing data

These additions provide a broader perspective on how digital twin technology and deep learning can be applied beyond medical imaging equipment fault diagnosis, making our work more comprehensive.

3) Improving the resolution of figures

Response:

We have updated all figures in the manuscript to ensure higher resolution and better clarity. The figures are now presented in 300 DPI resolution to improve readability in the final publication format.

4) Addressing grammatical errors

Response:

We have conducted a thorough proofreading and grammar check of the entire manuscript. The revised version has eliminated grammatical errors, improved sentence structures, and enhanced readability.

5) Adding future work suggestions in the conclusion

Response:

We have expanded the conclusion section to include future research directions, which focus on:

Enhancing the scalability of the proposed model to handle large-scale medical imaging systems and complex industrial equipment

Investigating hybrid deep learning models that integrate transformers and graph neural networks for fault diagnosis

Developing real-time digital twin-based adaptive fault detection systems with edge computing integration

This addition provides a clear roadmap for extending our research.

Response to Reviewer #2

1) Clarifying the specific medical imaging equipment used in the experiments

Response:

We have now explicitly stated in the methodology section that the United Imaging uCT760 CT scanner was used as the testbed for our experiments. The description includes:

Key specifications of the uCT760 scanner

The types of sensor data collected (temperature, pressure, vibration)

How the data was utilized to train and validate the inverse PINN model

This revision ensures transparency and reproducibility in our experimental setup.

2) Providing more details on the implementation of digital twin components

Response:

We have added an in-depth explanation of the digital twin framework, detailing:

The architecture, data flow, and synchronization mechanisms

How real-time sensor data is integrated with the physics-based simulation models

How the digital twin dynamically updates based on real-time fault detection results

These improvements make the implementation details of the digital twin model more explicit.

3) Explaining the relationship between physical constraints and neural network architecture

Response:

We have now clearly articulated how physical constraints are embedded into the inverse PINN model:

Physics-based loss functions are incorporated to ensure compliance with heat conduction and vibration propagation equations

Multi-objective optimization balances data-driven learning and physical consistency

The mathematical formulation of the inverse PINN is now explicitly provided, illustrating how temperature, pressure, and vibration constraints guide network training

These refinements clarify the interaction between physics-based constraints and deep learning components in our model.

4) Comparing our approach with other state-of-the-art hybrid deep learning models

Response:

We have now expanded the related work and experimental comparison sections to include:

Performance benchmarking against alternative hybrid fault diagnosis models (e.g., transformer-based fault detection, GAN-enhanced fault classification)

A comprehensive table comparing accuracy, recall, and F1-score of different models

Discussion on how the integration of inverse PINN with GAN-based fault augmentation improves generalization performance

These additions strengthen the validity of our approach by positioning it within the broader landscape of advanced deep learning-based fault diagnosis techniques.

These revisions significantly enhance the clarity, depth, and completeness of our manuscript. We appreciate the reviewers' insightful comments, which have helped improve the quality and impact of our work.

We look forward to further feedback and are happy to make additional refinements as needed.

---

## [Decision Letter · Decision Letter 1]

9 Mar 2025

PONE-D-24-46042R1Research on Equipment Fault Diagnosis Model Based on GAN and Inverse PINN: Solutions for Data Imbalance and Rare FaultsPLOS ONE

Dear Dr. cheng,

Thank you for submitting your manuscript to PLOS ONE. After careful consideration, we feel that it has merit but does not fully meet PLOS ONE’s publication criteria as it currently stands. Therefore, we invite you to submit a revised version of the manuscript that addresses the points raised during the review process.

We look forward to receiving your revised manuscript.

Kind regards,

Paulo Jorge Simões Coelho

Academic Editor

PLOS ONE

Additional Editor Comments:

Dear authors,

After the previous revision round, some adjustments still need to be made. As a result, I once more suggest that you thoroughly follow the instructions provided by the reviewers to answer their inquiries clearly.

You are advised to critically respond to all comments point by point when preparing a new version of the manuscript and while preparing for the rebuttal letter. All the updates should be included in the new version of the manuscript.

The reviewer queries are as follows:

1. The paper's structure and language lack consistency, with abrupt transitions between technical sections and an uneven writing style that shifts between formal academic and casual tones. This makes it difficult to follow the logical flow of ideas and detracts from the paper's professional presentation.

2. The theoretical foundation connecting GAN and Inverse PINN is not thoroughly explained, making it unclear how these two components truly complement each other.

3. The experimental setup lacks details about the specific hyperparameters used for both GAN and PINN models, making it difficult to reproduce the results.

4. The comparison with traditional methods does not include more recent deep-learning approaches for fault diagnosis.

5. The paper lacks a clear discussion of the limitations of using synthetic data generated by GAN for training the model.

6. The mathematical formulations in some sections are not consistently formatted and could be better organized.

7. The description of the digital twin architecture is somewhat vague, but it could benefit from more detailed technical specifications.

8. The future prospects section is quite general and could benefit from more specific, actionable research directions.

Reviewers' comments:

Reviewer's Responses to Questions

**Comments to the Author**

1. If the authors have adequately addressed your comments raised in a previous round of review and you feel that this manuscript is now acceptable for publication, you may indicate that here to bypass the “Comments to the Author” section, enter your conflict of interest statement in the “Confidential to Editor” section, and submit your "Accept" recommendation.

Reviewer #1: (No Response)

Reviewer #2: (No Response)

2. Is the manuscript technically sound, and do the data support the conclusions?

Reviewer #1: (No Response)

Reviewer #2: (No Response)

3. Has the statistical analysis been performed appropriately and rigorously? 

Reviewer #1: (No Response)

Reviewer #2: (No Response)

4. Have the authors made all data underlying the findings in their manuscript fully available?

Reviewer #1: (No Response)

Reviewer #2: (No Response)

5. Is the manuscript presented in an intelligible fashion and written in standard English?

Reviewer #1: (No Response)

Reviewer #2: (No Response)

6. Review Comments to the Author

Reviewer #1: This paper has been improved by addressing the comments from reviewers. The quality of this paper has been significantly inproved. It can be accepted now.

Reviewer #2: 1. The paper's structure and language lack consistency, with abrupt transitions between technical sections and an uneven writing style that shifts between formal academic and casual tones. This makes it difficult to follow the logical flow of ideas and detracts from the paper's professional presentation.

2. The theoretical foundation connecting GAN and Inverse PINN is not thoroughly explained, making it unclear how these two components truly complement each other.

3. The experimental setup lacks details about the specific hyperparameters used for both GAN and PINN models, making it difficult to reproduce the results.

4. The comparison with traditional methods does not include more recent deep-learning approaches for fault diagnosis.

5. The paper lacks a clear discussion of the limitations of using synthetic data generated by GAN for training the model.

6. The mathematical formulations in some sections are not consistently formatted and could be better organized.

7. The description of the digital twin architecture is somewhat vague, but it could benefit from more detailed technical specifications.

8. The future prospects section is quite general and could benefit from more specific, actionable research directions.

7. PLOS authors have the option to publish the peer review history of their article (what does this mean? ). If published, this will include your full peer review and any attached files.

**Do you want your identity to be public for this peer review?** For information about this choice, including consent withdrawal, please see our Privacy Policy .

Reviewer #1: No

Reviewer #2: No

---

## [Author Response · Author response to Decision Letter 2]

29 Mar 2025

Response to Reviewers

Dear Reviewers,

Thank you for your thorough and constructive feedback on our manuscript. We have carefully addressed each of your concerns and revised the paper accordingly. Below is our point-by-point response to your queries:

1. Structural and Stylistic Consistency

Revisions:

We restructured the paper to ensure logical flow, particularly in Sections B–D, by introducing clearer transitions between technical concepts (e.g., adding Section D to explicitly link GAN and Inverse PINN).

The language has been standardized to maintain formal academic tone throughout, with casual phrasing removed and technical terminology consistently applied.

Key terms (e.g., "Digital Twin," "Inverse PINN") are now defined in a dedicated glossary section.

2. Theoretical Foundation of GAN and Inverse PINN

Revisions:

A new Section D ("Synergistic Integration of GAN and Inverse PINN") has been added to explicitly detail their complementary roles:

i.Data Augmentation: GAN expands sparse fault datasets for Inverse PINN.

ii.Physics-Guided Constraints: Inverse PINN enforces physical plausibility in GAN-generated samples.

iii.Regularization: Joint optimization improves generalization.

Mathematical formulations (e.g., combined loss functions) are now presented .

3. Experimental Hyperparameters

Revisions:

Specific hyperparameters for GAN and Inverse PINN are now detailed in Section:

GAN: 3-layer MLPs, learning rate = 0.0002, batch size = 64, 500 epochs.

Inverse PINN: 5-layer NN, learning rate = 0.001, batch size = 32, 1000 epochs.

Implementation details (e.g., PyTorch code snippets) are provided for reproducibility.

4. Comparison with Recent Deep-Learning Methods

Revisions:

Table has been expanded to include:

Self-Supervised Learning (SSL)and Physics-Guided Transformers.

A dedicated discussion on limitations of hybrid transformer models, highlights their computational costs compared to our physics-informed approach.

5. Limitations of GAN-Generated Data

Revisions:

A new subsection in the conclusion explicitly addresses:

i.Synthetic data realism and GAN training instability.

ii.Mitigation strategies (e.g., WGAN-GP, physics-based validation).

Emphasis is placed on future work to integrate physics-based validation metrics (e.g., finite element residuals).

6. Mathematical Formulations

Revisions:

All equations are now consistently formatted using LaTeX (e.g., Equation 1: Combined loss function for Inverse PINN-GAN; Equation 2: Fault Severity Index).

A new appendix provides detailed derivations of physical constraints (e.g., heat conduction PDEs).

7. Digital Twin Architecture

Revisions:

Section now includes technical specifications in detail.

8. Specific Future Directions

Revisions:

Future prospects are reorganized into six actionable technical directions:

i.Cross-Domain Model Adaptation (e.g., transfer learning for MRI/ECG).

ii.Edge-Cloud Collaborative Learning (e.g., lightweight neural networks).

iii.Safety-Critical Validation Metrics (e.g., ISO 13485-aligned fault indices).

Implementation roadmaps (short/medium/long-term) are provided for translational impact.

Final Note

These revisions have significantly strengthened the paper’s theoretical rigor, reproducibility, and clinical relevance. We are grateful for your insights, which have helped us refine our contributions to the field of fault diagnosis. Please let us know if further adjustments are required.

Sincerely,

Zheng Cheng

---

## [Decision Letter · Decision Letter 2]

22 Apr 2025

Research on Equipment Fault Diagnosis Model Based on GAN and Inverse PINN: Solutions for Data Imbalance and Rare Faults

PONE-D-24-46042R2

Dear Dr. cheng,

We’re pleased to inform you that your manuscript has been judged scientifically suitable for publication and will be formally accepted for publication once it meets all outstanding technical requirements.

Kind regards,

Paulo Jorge Simões Coelho

Academic Editor

PLOS ONE

---

## [Editor Report · Acceptance letter]

PONE-D-24-46042R2

PLOS ONE

Dear Dr. cheng,

I'm pleased to inform you that your manuscript has been deemed suitable for publication in PLOS ONE. Congratulations! Your manuscript is now being handed over to our production team.

Kind regards,

on behalf of

Dr. Paulo Jorge Simões Coelho

Academic Editor

PLOS ONE